# Genome-Scale Metabolic Reconstruction and *in Silico* Perturbation Analysis of the Polar Diatom *Fragilariopsis cylindrus* Predicts High Metabolic Robustness

**DOI:** 10.3390/biology9020030

**Published:** 2020-02-17

**Authors:** Michel Lavoie, Blanche Saint-Béat, Jan Strauss, Sébastien Guérin, Antoine Allard, Simon V. Hardy, Angela Falciatore, Johann Lavaud

**Affiliations:** 1Unité Mixte Internationale 3376 Takuvik, CNRS-ULaval, Département de Biologie and Québec-Océan, Université Laval, Québec, QC G1V 0A6, Canada; blanche.saintbeat@gmail.com (B.S.-B.); sebastien.guerin@takuvik.ulaval.ca (S.G.); Johann.Lavaud@bio.ulaval.ca (J.L.); 2Department of Biology, University of Hamburg, D-22607 Hamburg, Germany; jan.strauss@cssb-hamburg.de; 3CSSB Centre for Structural Systems Biology, c/o Deutsches Elektronen-Synchrotron DESY, Notkestraße 85, D-22607 Hamburg, Germany; 4Département de physique, de génie physique et d’optique, Université Laval, Québec, QC G1V 0A6, Canada; Antoine.Allard@phy.ulaval.ca; 5Centre interdisciplinaire de modélisation mathématique, Université Laval, Québec, QC G1V 0A6, Canada; 6Département d’informatique et génie logiciel, Département de biochimie, microbiologie et bio-informatique, Université Laval, Québec, QC G1V 0A6, Canada; simon.hardy@ift.ulaval.ca; 7Unité des Neurosciences cellulaires et moléculaires, Centre de recherche CERVO, Québec, QC G1V 0A6, Canada; 8Institut de Biologie Physico-Chimique, Laboratory of Chloroplast Biology and Light Sensing in Microalgae, UMR7141, Centre National de la Recherche Scientifique (CNRS), Sorbonne Université, 75005 Paris, France; angela.falciatore@upmc.fr

**Keywords:** flux balance analysis, metabolic network, arctic, systems biology, reaction deletion, gene deletion

## Abstract

Diatoms are major primary producers in polar environments where they can actively grow under extremely variable conditions. Integrative modeling using a genome-scale model (GSM) is a powerful approach to decipher the complex interactions between components of diatom metabolism and can provide insights into metabolic mechanisms underlying their evolutionary success in polar ecosystems. We developed the first GSM for a polar diatom, *Fragilariopsis cylindrus*, which enabled us to study its metabolic robustness using sensitivity analysis. We find that the predicted growth rate was robust to changes in all model parameters (i.e., cell biochemical composition) except the carbon uptake rate. Constraints on total cellular carbon buffer the effect of changes in the input parameters on reaction fluxes and growth rate. We also show that single reaction deletion of 20% to 32% of active (nonzero flux) reactions and single gene deletion of 44% to 55% of genes associated with active reactions affected the growth rate, as well as the production fluxes of total protein, lipid, carbohydrate, DNA, RNA, and pigments by less than 1%, which was due to the activation of compensatory reactions (e.g., analogous enzymes and alternative pathways) with more highly connected metabolites involved in the reactions that were robust to deletion. Interestingly, including highly divergent alleles unique for *F. cylindrus* increased its metabolic robustness to cellular perturbations even more. Overall, our results underscore the high robustness of metabolism in *F. cylindrus*, a feature that likely helps to maintain cell homeostasis under polar conditions.

## 1. Introduction

Diatoms often dominate in the water column and sea ice of polar habitats, which exhibit large seasonal changes in temperature and light [1,2]. They can grow at temperatures near or below the freezing point, as well as under very low irradiance in and under ice, and under high irradiance in summer [3,4]. Living in the polar seas requires specific adaptations and acclimation strategies including cold-adapted enzymes, changes in membrane lipid composition, and antifreeze proteins [5,6,7]. Polar diatoms also have a higher protein content (e.g., rubisco) than temperate diatoms, which allow them to partially compensate for the low catalytic rate at low temperature [8,9], and they are enriched in polyunsaturated fatty acids and carotenoids [7,10]. Polar diatoms generally grow at lower rates than their temperate counterparts, and they exhibit low light saturation parameters for photosynthesis [11].

Although our knowledge on the peculiar physiology and metabolism of polar diatoms has made significant progress [3,4], we cannot fully explain their ecological success in cold environments. The use of recently developed integrative systems biology methods, based on the genome-scale metabolic model (GSM), can integrate interrelationships among cellular components and underscore emergent metabolic properties. Flux balance analysis (FBA) allows for the estimation of optimal steady-state cellular fluxes and growth through all biochemical and transport reactions given only a few experimentally measured metabolic constraints and knowledge of the molecular network (i.e., GSM) [12,13]. Such an approach highlights the way reactions interact with each other, and it can propose new metabolic pathways, like, for instance, the glutamine-ornithine shunt in the model temperate diatom *Phaeodactylum tricornutum* [14]. The use of FBA can also provide new insights into how metabolism can be regulated as a function of nutrients or light availability, as recently applied in *P. tricornutum* [14,15,16,17]. However, to date, little is known on the capacity of diatom metabolic networks at the omic-level to maintain cell homeostasis in response to environmental perturbations. While the metabolic network robustness in yeast, bacteria, and eukaryotic organisms other than microalgae has been intensely studied [18,19,20,21], only a few studies have investigated the robustness of cell metabolism (i.e., the capacity of the cell to resist changes in model parameters) of temperate microalgae [15,22]. It has been shown that genome-scale metabolic modeling of metabolism in temperate microalgae yields robust growth rates by changing most model parameters [15,22]. The FBA modeling approach at the genome scale allows determining the parameters controlling microalgae cell growth and defining the degree at which whole cell metabolism is sensitive to metabolic perturbations but has not been applied to a polar diatom model. Such a work is now possible using recent genomic and physiological data from a range of environmental conditions in the model polar diatom *Fragilariopsis cylindrus* [23,24].

A special feature of the 61.1 megabase’s draft genome of *F. cylindrus* is that it contains 6071 highly divergent alleles (sequence divergence of up to 6%) out of a total of 21,066 predicted genes [23]. Interestingly, the highly divergent alleles were differentially expressed across environmental conditions and suggested to be involved in the adaptation to strong environmental fluctuations in the Southern Ocean [23,25]. Reconstruction of a genome-scale model from these genomic data will help to estimate to what extent the metabolism and growth of a common polar diatom [23,26] is robust to cellular perturbations, a feature that likely helps diatoms to thrive under extreme and variable polar conditions.

In this study, we developed a genome-scale metabolic model of *F. cylindrus* including its special feature of highly divergent alleles, and we analyzed the robustness of algal growth to both perturbations of cellular biochemical composition and metabolic network structure.

## 2. Methods

### 2.1. Metabolic Network Reconstruction

We used a GSM of *Phaeodactylum tricornutum* (*i*LB1025), which was developed by Levering et al. [14] and updated by Broddrick et al. [15], as a template for the metabolic model of *Fragilariopsis cylindrus* (CCMP1102). BlastP analyses revealed that 93.6% of proteins in the *P. tricornutum* model (see Appendix A for a list of the proteins for which we did not find significant homology in the *F. cylindrus* genome) have significant homologous proteins in the *F. cylindrus* genome using an E cut-off <0.1 and overall identity >20%. In 39% (26 protein IDs and their associated protein names from the list shown in Appendix A) of cases when we did not identify significant homologous proteins, it was due to the presence of incomplete sequences caused by genome sequencing gaps in either the *P. tricornutum* or *F. cylindrus* genome sequence. We did not include these incomplete gene models and the 61% of genes without any significant blast hit and their associated protein IDs in our model. However, we opted to keep all cellular reactions (enzymatic reactions, transport, etc.) from the *P. tricornutum* model based on the general high similarity between proteins of *P. tricornutum* and *F. cylindrus* and included reactions for which no significant homologous genes were detected without any gene sequence association to provide a first robust reconstruction of a metabolic model for *F. cylindrus*. Additionally, subcellular locations of homologous proteins in *F. cylindrus* were predicted using web-based software (ChloroP, MitoProt, and DeepLoc-1.0) and shown to be identical for 96.6% of proteins in *P. tricornutum* (see Appendix A for a list of the proteins for which subcellular location of homologous proteins was misclassified). We acknowledge that the *F. cylindrus* cellular reactions included in our metabolic model are an approximation based on a temperate diatom species, but we stress that we focused on emergent metabolic network property such as metabolic robustness, rather than on specific fluxes for a given biochemical pathway. We updated the protein IDs of each cellular reaction for *F. cylindrus* based on manual protein annotations available on the Joint Genome Institute website (https://genome.jgi.doe.gov/pages/search-for-genes.jsf?organism=Fracy1) and on our BlastP analyses. We also added the protein IDs of 53 allelic variants corresponding to protein IDs that were already included in the model (see Appendix A for a list of the 53 protein IDs and their associated 111 enzyme names; i.e., some protein IDs are annotated with more than 1 enzyme name). The divergent allelic variants were resolved and validated using independent sequencing approaches and genome assembly approaches [25].

The reconstructed *F. cylindrus* metabolic model contains 2144 reactions, 1707 metabolites, 830 genes, and 1650 gene product relationships distributed across 6 compartments (cytosol, mitochondria, chloroplast, thylakoids, peroxisome, and extracellular medium), and 88 subsystems (see file model_setup.m). The model was constrained with 69 parameters based on experimental data for *F. cylindrus* performed in our lab and data available from previous studies (Table 1). When no data were available for *F. cylindrus*, data measured in *P. tricornutum* were used (Table 1 and Appendix A), bearing in mind that sensitivity of the model to these parameters was further evaluated. The model included 66 parameters describing the biochemical composition of a cell, which were used to determine stoichiometric coefficients of reactions in the biomass reactions of the model. It also included 3 parameters (C uptake rate, C:N and N:P total molar ratio) constraining nutrient uptake (Table 1 and Appendix A). Parameter values for C uptake rate, C:N, N:P ratio, pigment, and nucleic acid base content were measured or calculated for *F. cylindrus* cultures in controlled conditions (Appendix A). Note that we assumed that HCO_3_^-^ was the dominant inorganic carbon source taken up by *F. cylindrus,* as in the *P. tricornutum* GSM. Other model parameters lacking for *F. cylindrus* were taken from other temperate marine diatoms than *P. tricornutum* based on available literature data, and the sensitivity of each parameter was further tested, as described below. Details on *F. cylindrus* culturing and experimental analysis, as well as the source of the other parameter values, are described in Appendix A. Mean values of experimentally determined physiological data were used as model parameters.

The model assumes non-growth-limiting nutrient and light conditions. Therefore, the model only takes into account scenarios of light wavelength or intensity that lead to optimal growth and the influence of light wavelength and intensity on photosynthesis efficiency and pigment synthesis is not considered. Such a work would necessitate the coupling of our genome-scale model to a physiological model relating light and algal metabolism. Moreover, the model does not account for regulatory mechanisms such as feedback interactions, allosteric enzyme activation, or transcription regulation; the model rather relies on the metabolic optimality hypothesis, which is discussed in the next section, as a way to gain insights on the behavior and robustness of *F. cylindrus* metabolism.

### 2.2. Flux Balance Analysis (FBA)

We used FBA to compute fluxes of all reactions in the model. FBA is a mathematical method for underdetermined equation systems and can thus yield multiple possible solutions. Optimization through the use of objective functions is necessary to determine a single (unique or nonunique) flux distribution. Here, we used an objective function that first maximizes biomass production, since *F. cylindrus* cells can divide during the dark and the light phases of the diel cycle with a secondary objective of minimizing the Manhattan norm of the flux vector representing the cell’s strategy to minimize the sum of flux values. This dual objective function yields a unique flux distribution when performing FBA [14,15,17]. It assumes that cell metabolism and enzyme usage tend toward optimality, and it means that algal growth is maximized per unit of flux. Intracellular flux distribution measured using ^13^C and growth of the prokaryotic organisms *Escherichia coli* [29,30,31], the yeast *Saccharomyces cerevisiae* [32], and the cyanobacterium *Synechococcus elongatus* [33] in optimal culture conditions has indeed been shown to be well-predicted by parsimonious FBA, which maximizes growth per unit of flux. However, to our knowledge, no reports specifically address this hypothesis in eukaryotic microalgae. In those microalgae, nonadaptative forces such as genetic drifts or mutations may well be important evolutionary drivers of metabolism (Lynch, 2007), in addition to natural selection, which selects optimal fitness over time, but the relative quantitative importance of both processes remains unknown. Even though we acknowledge that futile cycles and metabolic constraints can lead to some nonoptimal reactions (e.g., cyclic electron transport or proton slippage in electron transport chains) [34], our assumption of general metabolic optimality, particularly in conditions maximizing the growth rate in *F. cylindrus*, is necessary for efficiently solving a complex underdetermined system of cellular reactions and is, in our view, reasonable as a first-order estimate.

### 2.3. Quality Control of the Genome-Scale Model

Network evaluation of proper behavior and quality assurance was evaluated following standard methods (see file quality_control.m) [35,36]. The capability of the model to produce biomass precursors was validated, and the model was tested for proper behavior when key reactions were deleted, as assessed through visualization of the network with the Paint4Net Cobra Toolbox [37]. We validated that there were no stoichiometric errors in the metabolic network leading to ATP and NADP production without input of carbon, nutrient, and light. This quality control was done by setting carbon and light inputs to zero and performing FBA or by considering no exchange reactions while optimizing the flux through the ATP maintenance reaction or through an artificial reaction (NADPH → NADP + H^+^) and performing FBA. Since no FBA solutions were obtained or all fluxes were 0, it was therefore verified that ATP and NADP could not be produced from nothing. Leak tests also show that no metabolite is produced when all exchange reactions are closed.

### 2.4. Sensitivity Analysis of Model Output

Sensitivity analyses of the FBA output were performed for parameters, reactions, and genes. For the parameter sensitivity analysis, we used local and global sensitivity analyses. Local sensitivity analysis evaluates the sensitivity of the growth rate to single parameter changes, while global sensitivity analysis gives information on the sensitivity of each parameter and their relative interactions. The parameter sensitivity analysis evaluates the robustness of the model. Regarding the sensitivity analysis on reactions or genes, we performed single reaction or gene deletion in order to produce a model with a modified structure (MMS) and compared it to the full initial model (IM or control). This sensitivity analysis on reaction or gene evaluates model structural robustness. In this case, we computed growth rates for the MMS and the IM using FBA and quadratic minimization of the metabolic adjustment (MOMA). The evaluation of gene structural robustness was performed with and without inclusion of divergent allelic variants to further test the hypothesis that the highly divergent alleles in *F. cylindrus* increase metabolic robustness to cellular perturbations. Note that the MOMA algorithm looks for an optimal flux distribution, as done by FBA, while also minimizing metabolic changes for acclimation in the MMS relative to the control (IM). It therefore better reflects the tendency of biological systems to buffer cellular perturbations on a short-term period, while FBA predicts the final steady-state distribution of fluxes, including potential long-term acclimation responses.

### 2.5. Model Implementation in MATLAB

Model development, sensitivity analyses, and flux distribution calculations were performed using MATLAB 2018 (The MathWorks Inc., Natick, MA, USA). All custom codes are available on GitHub (https://github.com/michellavoie4/Resilience-Fcylindrus). The scripts for the local and global sensitivity analysis to changes in the 69 parameters of the model are detailed in the files localsens_script.m and globalSA_script.m using the functions localsensFBA.m and FBAlight.m. The global sensitivity analysis was done using the Morris method, which is recommended for models with a high number of parameters. When running the Morris method, we used the Sensitivity Analysis For Everyone (SAFE) toolbox in MATLAB, taking into account the latin hypercube resampling strategy and 6500 model evaluations [38].

Sensitivity of growth to reaction or gene deletion was assessed with the Constraint-Based Reconstruction and Analysis (COBRA) toolbox functions *singleRxnDeletion()* or *singleGeneDeletion()* using FBA or MOMA (see the file ReactionDeletionSensitivity.m) with the prior addition of constraints on photon uptake (i.e., allowing a 1.5-fold increase or decrease with respect to the control without reaction deletions) to avoid modeling unrealistically high light harvesting in response to deletions. We set the computed growth rate to 0 when reaction or gene deletions lead to infeasible problems.

FBA and MOMA were performed using the Gurobi Optimizer Version 7.5.2 (Gurobi Optimization Inc., Houston, TX, USA) solver (http://www.gurobi.com) with the Constraint-Based Reconstruction and Analysis (COBRA) toolbox v3.0 [36]. The minimum nonzero fluxes obtained after FBA or MOMA (i.e., detection limit of fluxes) with our computer system configuration is 10^−6^ for the reaction of the objective function and 10^−9^ for all other reactions in the model.

### 2.6. Network Theory Analysis

We extended the analysis of structural robustness of the network based on single reaction deletions by additionally calculating connectivity and centrality indices of metabolites involved in active reactions. Therefore, the COBRA metabolic network (a directed bipartite network) was exported to Python 3 [39] for further analyses (see Appendix A for further details and GitHub files: bipartite.m, main_network_analysis.py, and network_metrics.py) using the package *Networkx* [40]. The general structure of the metabolic model in Python, which consists of two types of nodes (i.e., metabolites or reactions), was the same than that of the COBRA model. However, we only considered active reactions (i.e., around a quarter of all reactions) in the Python bipartite network, since we aimed to study the effect of deletion of active reactions on model properties. The python network contains 1199 active nodes, including 627 metabolites and 572 reactions. All substrates (i.e., metabolites used in a reaction) are linked to their respective reactions, which are then linked to products (metabolites produced by a reaction).

We considered the three connectivity indices total degree, in-degree, and out-degree. When considering metabolite nodes only, “total degree” describes the total number of reactions connected to a metabolite, “in-degree” the number of reactions producing a metabolite, and “out-degree” describes the number of reactions using a metabolite as a substrate. We also calculated the betweenness and closeness centrality indices. Betweenness centrality describes the number of shortest paths crossing a given node and is thus a proxy of influence a node has over the flow of matter in the network. Closeness centrality is the reciprocal of the sum of the length of all shortest paths between the node and all other nodes in the network and is thus a measure of closeness for a given node to all other nodes in the network.

## 3. Results and Discussions

We developed the first genome-scale model of the polar diatom *Fragilariopsis cylindrus* using available physiological and genomic data. The model allows to determine the robustness of the growth rate and metabolism to cellular perturbations, such as changes in concentration of biochemical components or inactivation of a given reaction or gene.

### 3.1. Prediction of Energy Dissipation Pathways, Cell Bioenergetics, and Growth Rate with Flux Balance Analysis (FBA)

An FBA using the exchange constraints and parameters listed in Table 1 predicts that 574 (27%) of all reactions in the model are active (have nonzero fluxes), while 1570 (73%) reactions are inactive. As expected under the optimality hypothesis of FBA, our model predicts that energy dissipation does not occur; i.e., photorespiration, glycolate production, and carbon exudation, as well as cyclic electron, flows through the Mehler reaction, and the plastid terminal oxidase-PTOX are inactive. No carbon loss occurred in the model because a small proportion (10%) of cellular CO_2_ (produced mostly from pyruvate dehydrogenase and HCO_3_^−^) is fixed as organic carbon (oxaloacetate) by a chloroplastic phosphoenolpyruvate carboxylase (JGI protein IDs: 259279), an enzyme of the C4 photosynthesis pathway which is incomplete in the model, and leads to amino acid synthesis (glutamine). The remaining part of the absorbed CO_2_ (90%) is fixed by Rubisco (JGI protein IDs: 184116 and (147482 or 196615 or 195439)).

FBA modeling also enabled us to decipher cell bioenergetics in *F. cylindrus*. Regarding the ATP:NADPH balance, we considered two scenarios: (1) no redox exchange between chloroplast and mitochondria or (2) a coupling of mitochondrial ATP generation to electron transfer at photosystem I (PSI; as implemented in the *P. tricornutum* model of Levering et al. [14] based on the findings of Bailleul et al. [41]). During the first scenario, without considering an exchange of redox compounds between the mitochondria and chloroplast, we found a PSI cyclic electron flow that led to the production of 0.27 mmol ATP gDW^−1^ h^−1^. This represents about 10% of the total ATP synthesized in the chloroplast by the ATP synthase. In comparison, for the second scenario, when coupling the mitochondrial ATP generation to electron transfer at PSI and setting alternative oxidase (AOX) flux equals to 0 (otherwise this reaction is activated and the mitochondrial electron transport chain is short-circuited), 0.27 mmol ATP gDW^−1^ h^−1^ were produced in the mitochondria rather than in the chloroplast. The model simulates efficient communication between both organelles and suggests that a small proportion of mitochondrial ATP gross production has to be synthesized to control the cellular ATP:NADPH ratio in *F. cylindrus* under optimal/suboptimal light conditions. Therefore, under both scenarios, mitochondrial ATP production through respiration contributes a small fraction of <13% towards the total cellular ATP production when growth is optimal.

The net gain of cellular energy management controls the algal growth rate. The predicted steady-state growth rate of *F. cylindrus* using FBA was between 0.37 and 0.46 d^−1^, considering a carbon uptake rate between 0.70 and 0.85 mmol C mg DW^−1^ h^−1^. This carbon uptake rate is based on our own laboratory measurements of short-term ^14^C uptake rates in optimally growing *F. cylindrus* cells (1.04 mg C mg Chl *a*^−1^ h^−1^ ± (SD) 0.12) (Appendix A). This modeled growth rate is close to the mean experimentally measured steady-state growth rate of 0.30 ± (SD) 0.05 d^−1^ determined in the same cultures as described in Appendix A. It suggests that the synthesis of uncharacterized osmolytes in diatoms, which are not accounted for in the model and could have lowered the growth rate more substantially, do not represent a high energy cost, as predicted for small diatoms [42]. It also indicates that energy dissipation mechanisms (e.g., cyclic electron flow or carbon exudation), which are predicted to play a minor role in our model and could potentially decrease the modeled growth rate, do not affect the net carbon uptake under these optimal growth conditions. Although a substantial proportion of electrons can be diverted to cyclic electron flow in *F. cylindrus* when grown in continuous high light at 150 µmol photons m^−2^ s^−1^ [43], the similarity of our modeled and measured growth rates suggest that cyclic electron flow is negligible at lower light intensity.

### 3.2. Sensitivity Analysis of Model Parameters

Local sensitivity analysis showed that modeled growth rates were robust (i.e., varied by less than 4.2%) to an arbitrary selected (but quantitatively significant) 40% increase or decrease in 68 out of the 69 parameters independently. Essentially, only carbon uptake rate controls the growth rate. More specifically, the growth rate increased by 40% or decreased by 29% in response to a 40% increase or decrease in carbon uptake rate, respectively (Figure 1). Most sensitive model parameters after the carbon uptake rate were total protein, carbohydrate, and lipid cellular concentrations, as well as total C:N or N:P ratios (Figure 1 and Figure 2). These were also the parameters with the strongest interactions with other parameters based on a global sensitivity analysis (Figure 2). The fluxes through equations of total pigments, total lipids, total carbohydrates, total DNA, total RNA, total proteins, and carbon storage (i.e., glucan and triacylglycerol) were also not disproportionately sensitive to changes in model parameters (i.e., <40.4% change in all cases), indicating that the synthesis of various fractions of cellular carbon was not less robust to uncertainty around model parameters than the total carbon uptake rate (see Appendix A, Appendix A).

To assess the strength of potential interactions among parameters, a local sensitivity analysis with 100,000 combinations of random parameters, which varied by 40%, was initially performed. The maximum relative increase or decrease in growth rate for multiple random local sensitivity analyses was 47.1% and 48.5%, respectively, suggesting that no strong synergistic interactions occur among parameters. Next, we calculated the relative error on growth rate due to changes by a factor of 1.4 of all combinations of the nine most sensitive parameters determined during our initial local sensitivity analysis (in decreasing order of sensitivity: carbon uptake rate, N:P ratio, C:N ratio, proportion of glucan, total lipids, total carbohydrates, total proteins, phenylalanine concentration, and aspartate concentration) and observed that the growth rate increased or decreased by less than 48.9% and 48.3%, suggesting that the interactions among parameters is quantitatively small, underscoring the robustness of *F. cylindrus* growth to changes in model parameters. Our results highlight the need to accurately measure and constraint the carbon uptake rate, which is by far the most sensitive parameter when modeling microalgal growth rate with FBA. Simultaneously, we show that reliable modeling of the growth rate can be achieved for *F. cylindrus* even under uncertainties about parameters that are currently not well constrained. It indicates that the reconstructed model can support suitable conclusions regarding the growth of *F. cylindrus*.

The strong robustness of the *F. cylindrus* growth rate to changes in concentrations of main cellular components (i.e., total pigments, total lipids, total carbohydrates, total DNA, total RNA, total proteins, and carbon storage) can be explained by two reasons. First, the sum of all biosynthetic rates of main cellular components must equal the fixed carbon uptake rate, and hence, a given relative change on one parameter will be compensated for by other components. Secondly, parameters of the model are cellular concentrations that are converted to stoichiometric coefficients, which are normalized with respect to the total mass per reaction in order to keep the reaction mass balance; this normalization also buffers the changes in the input parameters on reaction fluxes and growth rate. It is noteworthy that both explanations should generally also hold true for GSM developed for other species, suggesting that robustness of the growth rate and metabolism likely also occur in other (diatom) species. Using a coupled physiological-genome-scale model, Broddrick et al. [15] performed a sensitivity analysis of the modeled growth rate of *P. tricornutum* grown in a sinusoidal light regime to changes in the biomass composition, light intensity, and parameters related to cell physiology. They generated 1000 random combinations of parameters, for which a given percent of perturbations were applied (±0% to 50%) (see Figures S4 and S5 of Broddrick et al. [15]), finding that the percent change in the growth rate was smaller than or close to the perturbation of each component, supporting the general robustness of the diatom growth rates indicated in our study.

In Appendix A, we confirmed that the *P. tricornutum* modeled growth rate can be robust to changes in model parameters and compared the results to those found for the model parametrizes for *F. cylindrus*. A local sensitivity analysis using parameters for *P. tricornutum* yields similar trends to that for *F. cylindrus* (Figure 1), although the predicted growth rate for *P. tricornutum* was more sensitive to the use of a Redfield C:N:P ratio than to a low C:N and N:P ratio typical of polar algae. This is likely because a low C:N ratio lead to nitrate storage in the vacuoles through the use of a nitrate demand reaction (i.e., a reaction allowing unbalanced nitrate accumulation in the steady-state model), and this predicted excess in cell nitrogen could act as a buffer providing the required amount of nitrogen for optimal growth. To our knowledge, the presence and quantitative importance of a putative storage of cellular nitrogen in the vacuoles of *F. cylindrus* or *P. tricornutum* remains to be determined. Further experimental and modeling work is necessary in order to test the interesting hypothesis stating that storage of inorganic nitrogen in polar algae can favor growth rate robustness.

### 3.3. Effects of Single Reaction Deletions Calculated Using FBA and MOMA

In addition to testing the robustness of the *F. cylindrus* metabolic network to perturbations without changing the intrinsic model structure, we analyzed the robustness of the modeled growth rate and metabolism to structural modifications within the network (i.e., structural robustness of the metabolic network) ([19]). We found that a non-negligible proportion of reactions, i.e., 33% (192) and 20% (116 reactions) for the FBA and MOMA method, respectively, of all active reactions (Figure 3) can be deleted without affecting the growth rate by more than 1% (see Appendix A for a list of nonessential robust reactions). Taking also into account the 1570 nonactive reactions of the model discussed above, 79% (1686) to 82% (1762) of all reactions included in the model are dispensable in optimal growth conditions, again highlighting the robustness of metabolism in *F. cylindrus*. In comparison, single reaction deletions of 384 reactions (66%) and 396 reactions (68%) of active reactions abolished growth according to FBA and MOMA, respectively (see Appendix A for a list of essential sensitive reactions). These two groups of reactions were denoted “robust reactions” and “sensitive reactions”. In contrast to the almost total absence of intermediate growth inhibition predicted using the FBA method, an inhibition of the growth rate between 1% and 99% was calculated for ~12% (100 − 68 − 20 = 12) of independently deleted reactions using the MOMA method (Figure 3). This is consistent with a decrease in metabolic adjustments or robustness with the MOMA method compared to the FBA method.

Interestingly, when using either the FBA or MOMA methodology, after reaction deletions of robust reactions, the proportion of reactions being activated in the MMS in comparison to the control (IM) was always higher than the proportion of reactions being inactivated, suggesting that the robustness to reaction deletions is achieved through activation of compensatory reactions. This compensatory response was due to the presence of enzymes catalyzing the same reaction but using different energy sources (e.g., 5-methyltetrahydrofolate oxidoreductase using NAD or NADPH, both with the same JGI Protein ID of 225004, and methylenetetrahydrofolate dehydrogenase using NADP or NAD with JGI Protein IDs 196723 or 249281, respectively) or to enzymes catalyzing the same reaction in different cellular compartments, such as the cytosolic (JGI protein IDs 170472 or 209612), the mitochondrial (JGI Protein ID 193396), or the chloroplastic (JGI Protein ID 238714) aspartate aminotransferase, as well as the cytosolic (JGI Protein ID 236986) or mitochondrial (JGI Protein ID 246065) threonine aldolase with differential activation of energy sources or metabolite transporters. Fluxes through other reactions modulating the cell redox state (NADPH), such as ferredoxin-NADP reductase (H^+^ + NADP + 2 FDXRD < = > NADPH + 2 FDXOX, where FDXRD is reduced ferredoxin and FDXOX is oxidized ferredoxin) or thioredoxin reductase (NADPH + NAD < = > NADP + NADH), were also modulated to re-equilibrate the cell redox state. Among the energy dissipation pathways, photorespiration was only activated for two reaction deletions (glycine hydroxymethyltransferase with JGI Protein IDs of 239548 or 252263, as well as glycine synthase with JGI Protein IDs 271375, 224733, 275698, and 276117) in order to compensate the loss in glycine biosynthesis, but the Mehler reaction, as well as plastid terminal oxidase-PTOX, always remained inactive, according to FBA. In comparison, photorespiration was activated during deletion of 37% robust reactions, according to MOMA analysis. In all cases of robust reaction deletions (and predicted by FBA and MOMA methodology), the biosynthetic fluxes of proteins, lipids, carbohydrates, DNA, RNA, and pigments in cellular components varied by <0.1%. However, this metabolic flexibility entails a significant energy cost. According to FBA, photon harvesting and gross chloroplastic ATP and NADPH synthesis through the photosynthetic linear electron flow were consistently increased when deleting any of the robust reactions, while the cyclic electron flow at PSI and mitochondrial ATP production were activated when deleting 40% and 57% of the robust reactions, respectively. In comparison, according to MOMA analysis, photon harvesting and gross chloroplastic ATP and NADPH synthesis through the linear electron flow was constant but mitochondrial ATP production was activated when deleting any of the robust reactions, and the cyclic electron flow at PSI was activated during the deletion of 60% of the reactions.

In contrast to the deletion of individual robust reactions, more reactions were inactivated than activated when deleting sensitive reactions, according to both FBA and MOMA methodologies, leading to a lower total net number of active reactions for the MMS than for the IM. This is consistent with the “compensatory response” of structural robustness, as discussed above. The sensitive reactions are critical reactions in a given metabolic pathway for which no alternative route exists, such as the enzymes involved in synthesizing chrysolaminarin (e.g., Beta-1,3-Glucan synthase with a JGI Protein ID 261303), as well as N, P, and C uptake reactions; light harvesting; and biomass reactions leading to the synthesis of main cellular components (i.e., proteins, carbohydrates, lipids, pigments, DNA, and RNA). According to FBA, the compensatory responses in energy production were smaller when deleting sensitive reactions than when deleting robust reactions. Specifically, photon harvesting and gross chloroplastic ATP and NADPH synthesis through the linear electron flow was increased for 80% of sensitive reaction deletions, and mitochondrial ATP production was activated for 42% of sensitive reaction deletions. The cyclic electron flow around PSI, however, was activated for 46% of sensitive reaction deletions. According to the MOMA analysis, the cyclic electron flow at PSI was activated for 60% of single reaction deletions, and mitochondrial ATP production was always activated when deleting single sensitive reactions, as found when deleting robust reactions, but to a lesser extent. All the previously discussed contrasting cellular responses to single reaction deletions of robust or sensitive reactions are summarized in Table 2.

### 3.4. Analysis of Reaction Robustness Using Network Theory Metrics

The differences in structural robustness between reactions of the model can partly be explained by changes in degree number (i.e., reactions per metabolite) and centrality of metabolites involved in both reaction groups (Appendix A). Mean and median total degree per metabolite, as well as mean in-degree or out-degree, was about 50% higher for the metabolites involved in robust reaction deletion (“robust metabolites”) than for those involved in sensitive reactions (“sensitive metabolites”). This difference was essentially due to the lower proportion of metabolites involved in few reactions (i.e., poorly connected metabolites with in-degree or out-degree = 1 or total degree = 2), as well as a higher relative frequency of metabolites involved in a high number of reactions (high degree number) (e.g., ATP, proton, ADP, and water) (Figure 4 and Appendix A).

Similar results were found for the betweenness centrality indices; the mean and median betweenness centrality increased by 77% and 50% for the robust set of metabolites compared to the sensitive set of metabolites, respectively (Appendix A). The higher mean betweenness centrality of the robust set of metabolites was due to a lower proportion of metabolites with relatively low betweenness centrality (0 to 5000) (Appendix A). However, the closeness centrality index changed less than the betweenness centrality index between the two sets of metabolites. In contrast, the relative frequency at the lowest closeness centrality (0 to 0.005) for the robust metabolites decreased 1.5-fold relative to that of the sensitive metabolites, respectively (Appendix A). The presence of metabolites with higher betweenness centrality indices for robust metabolites compared to sensitive metabolites suggests that the reactions facilitating energy flow in the metabolic network promote the robustness of the network. This is expected under the hypothesis of metabolic optimality (i.e., maximization of biomass production and minimization of fluxes per unit of enzymes).

In brief, our analysis of metabolite centrality and degree number revealed that the network structural robustness increased as the proportion of metabolites involved in a low number of reactions decreased. Structural robustness was also linked to an increase in the proportion of high-degree metabolites involved in a high number of reactions (e.g., ATP, proton, ADP, and water). Our network theory analysis suggests that metabolic robustness in *F. cylindrus* was related to a flexibility in the choices of metabolites for a compensatory response by a given reaction. Moreover, our results have potential applications for the computational analysis of metabolic networks. We propose that the proportion of metabolites with low degree numbers can be used as indicators of network stability after reaction deletions. However, further modeling and experimental studies on other microorganisms are required to validate this potential network theory metric of robustness systematically.

### 3.5. Effects of Single Gene Deletion Calculated Using FBA and MOMA

Structural robustness at the genetic level was higher than at the reaction level; percentages of robust genes (affecting the growth rate by less than 1%) out of all active genes reached 42% and 29% for the FBA or MOMA method, respectively, without including highly divergent allelic variants (Figure 5A) compared to percentages of robust reactions of 33% or 20% for the FBA or MOMA method, respectively (Figure 3). This is because of the much larger number of reactions in the model (around 3.2-fold higher for the robust set of reactions) with alternative protein IDs catalyzing the same reaction (i.e., genes rules including the “OR” keyword) than the number of reactions with a set of coessential genes coding for enzyme subunits catalyzing one reaction (i.e., gene rules including the “AND” keyword). Strikingly, the inclusion of highly divergent allelic pairs that are distinct to *F. cylindrus* into our model further increased the predicted growth rate robustness of *F. cylindrus* to a single gene deletion analysis, with 55% or 44% of robust active genes for the FBA or MOMA method, respectively (Figure 5B), lending support to the hypothesis that the differentially expressed divergent alleles support growth under highly variable environmental conditions of the polar oceans [23]. When also considering the large proportion of protein IDs associated with inactive reactions (around 47%), we found that, in optimal growth conditions, a high proportion of *F. cylindrus* genes (70% to 76%) out of all included genes in the model are predicted to be dispensable’ i.e., that only 24% to 30% of genes are predicted to be essential. While studies on the structural robustness of the genetic network in microalgae are scarce, modeling and knock-out experiments in yeast, bacteria, and multicellular eukaryotes have also shown that only between 5% to 37% of all genes are essential at any given experimental conditions when considering one gene at a time [21]. In yeast, this vast pool of nonessential genes (and protein catalyzing reactions) is assumed to increase fitness under different environmental conditions by improving metabolic plasticity, as well as robustness, to mutations due to the presence of gene duplicates of different genes allowing flux reorganization [44,45]. Resolving the relative importance of both factors (mutational robustness and metabolic plasticity) explaining structural robustness of the *F. cylindrus* metabolic network will require further studies combining metabolic modeling and experimental gene/reaction deletion under different growth conditions.

## 4. Conclusions

We have derived the first genome-scale model for the polar diatom *Fragilariopsis cylindrus* and developed a MATLAB/Python workflow to analyze the sensitivity and robustness of the model. Our methodology is transferable to GSMs of other species, with some modifications in the code, and will help to improve the analysis of cellular metabolism in future studies using systems biology approaches. Specifically, our results highlight the high robustness of the *F. cylindrus* growth rate and metabolism to an array of cellular perturbations. Strikingly, our *in silico* predictions confirm that the highly divergent alleles that are unique to *F. cylindrus* increase metabolic robustness to cellular perturbations and likely contribute to the ecological success of *F. cylindrus* in polar environments. Since our model does not account for regulatory mechanisms, such as feedback interactions, allosteric enzyme activation, or transcription regulation, which enhance network stability to changes in the cell biochemical composition or reaction deletions, we speculate that the predicted model robustness may well be even more important for *F. cylindrus* than predicted by our genome-scale model. Our genome-scale model provides the basis to further study *F. cylindrus* metabolism constrained with physiological, metabolomic, and/or transcriptomic data. It unlocks new possibilities to decipher the intertwined relationships between environmental factors and microalgal metabolism, which will be likely of interest in theoretical and applied biology, as well as in microalgal-based biotechnology approaches.

## Figures and Tables

**Figure 1 biology-09-00030-f001:**
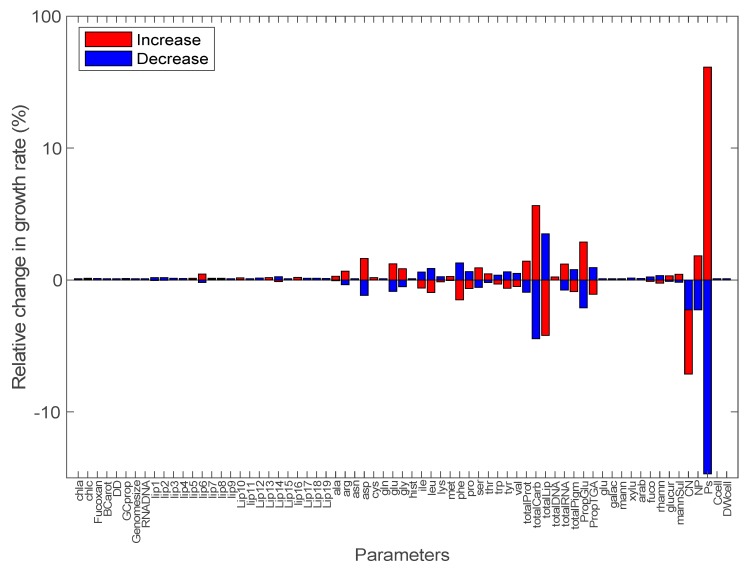
Local sensitivity analysis of the model output (growth rate) to an increase (red bars) or decrease (blue bars) by an arbitrary factor of 40% for each of the 69 model parameters taken independently. The percentage of variation of the growth rate is given on the y-axis (on a log scale) as a function of changes in parameter values (see Table 1 and Appendix A for details and units of parameters). (1) chla: chlorophyll a; (2) chlc: chlorophyll c; (3) Fucoxan: fucoxanthin; (4) BCarot: beta-carotene; (5) DD: diadinoxanthin; (6) GCprop: proportion of GC in DNA; (7) Genomesize: total genome size; (8) RNADNA: RNA:DNA ratio; (9–27) lip1 to lip19 refers to lipids tabulated in Appendix A; (28) ala: alanine; (29) arg: arginine; (30) asn: asparagine; (31) asp: aspartate; (32) cys: cysteine; (33) gln: glutamine; (34) glu: glutamate; (35) gly: glycine; (36) hist: histidine; (37) ile: isoleucine; (38) leu: leucine; (39) lys: lysine; (40) met: methionine; (41) phe: phenylalanine; (42) pro: proline; (43) ser: serine; (44) thr: threonine; (45) trp: tryptophane; (46) tyr: tyrosine; (47) val: valine; (48) totalProt: total protein; (49) totalCarb: total carbohydrate; (50) totalLip: total lipid; (51) totalDNA: total DNA; (52) totalRNA: total RNA; (53) totalPigm: total pigment; (54) PropGlu: proportion of glucan; (55) PropTGA: proportion of triacylglycerol; (56) glu: glucose; (57) galac: galactose; (58) mann: mannose; (59) xylu: xylulose; (60) arab: arabinose; (61) fuco: fucose; (62) rhamn: rhamnose; (63) glucur: glucuronate; (64) mannSul: mannose-sulfate; (65) CN: total C:N ratio; (66) NP: total N:P ratio; (67) Ps: C uptake rate; (68) Ccell: carbon per cell; and (69) DWcell: dry weight per cell.

**Figure 2 biology-09-00030-f002:**
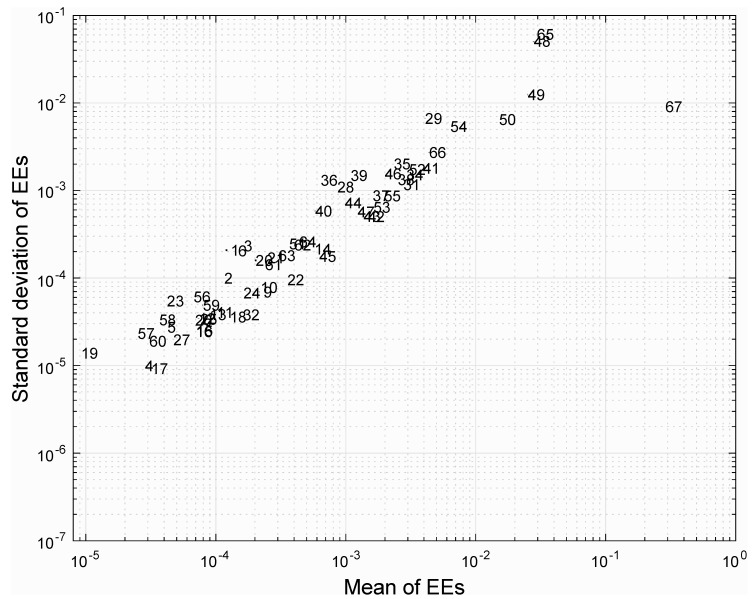
Global sensitivity analysis of the model output (growth rate) to a change by 40% of the 69 model parameters using the Morris method. The relative influence of each parameter on the model outcome (growth rate of *F. cylindrus* in the light) is quantified with a sensitivity index called “mean of elementary effect”, or mean EEs on the X-axis. The relative strength of interactions between parameters can also be visualized with the standard deviation of elementary effects (standard deviations of EEs) on the Y-axis. All points are numbered following the order of appearance of parameters in Figure 1 from left to right. Note that the parameters #7, #30, and #33 are not shown, since they have standard deviations of EEs and mean EEs smaller than 10^−11^.

**Figure 3 biology-09-00030-f003:**
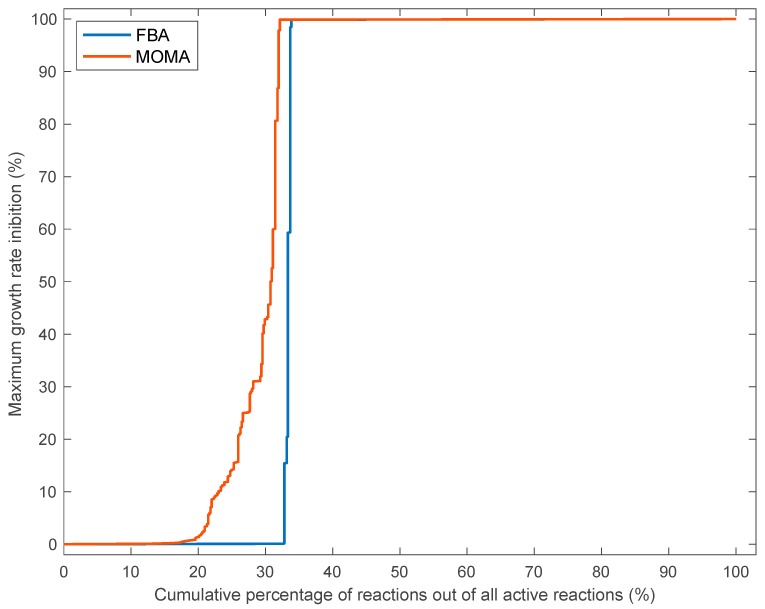
Cumulative percentage of active reactions, for which deletion of each reaction (taken independently) decreased the growth rate by a proportion equal to the value on the Y-axis or lower. Calculation of intracellular flux distribution and growth rate after single reaction deletion were performed using flux balance analyses (FBA, blue line) or quadratic minimization of metabolic adjustments (MOMA, red line). For instance, the curve describing the reaction deletion with FBA means that a single reaction deletion for around 15% of active reactions does not inhibit growth rate. For around 32% of all active reactions, single reaction deletion decreases growth rate by less than 1%. For around 68% (100−32) of all active reactions, single reaction deletion abolishes growth rate, so the cumulative sum on the X-axis gives 100. The other curves can be interpreted analogously.

**Figure 4 biology-09-00030-f004:**
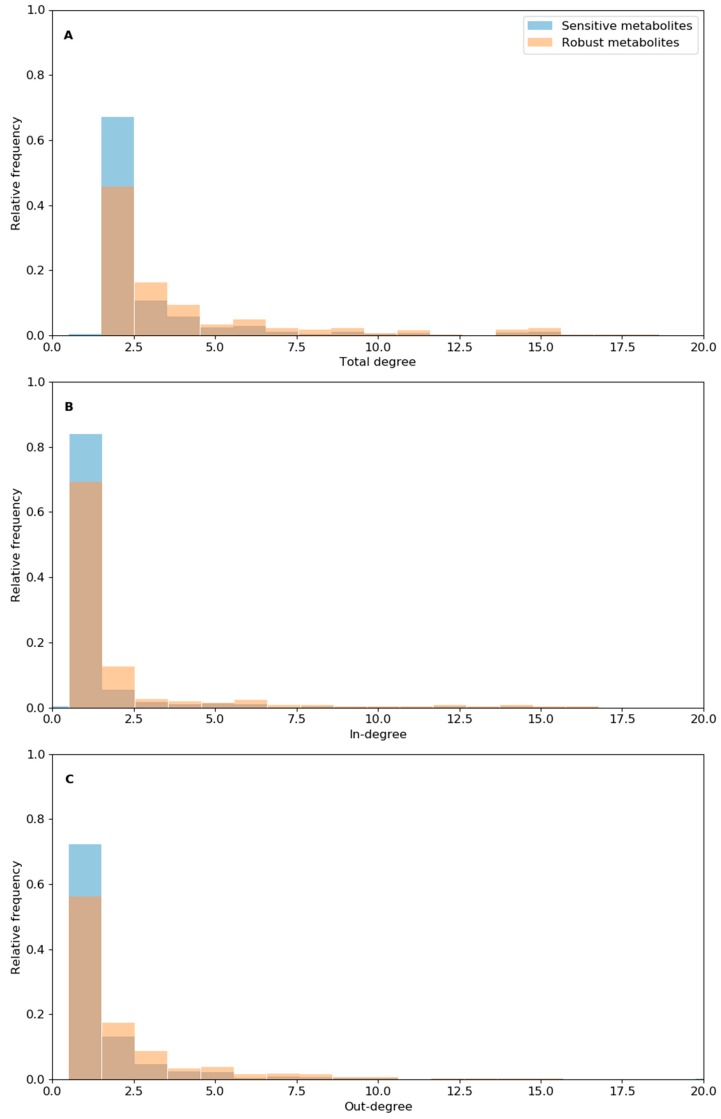
Relative frequency histogram of total degree (**A**), in-degree (**B**), and out-degree (**C**) for the metabolites involved in sensitive reactions (sensitive metabolites) or robust reactions (robust metabolites). “Total degree” refers to the total number of reactions connected per metabolite, “in-degree” refers to the number of reactions producing a metabolite, and “out-degree” is the number of reactions using a metabolite as the substrate. The X-axis is set to values smaller than 20. Full data are available in Appendix A.

**Figure 5 biology-09-00030-f005:**
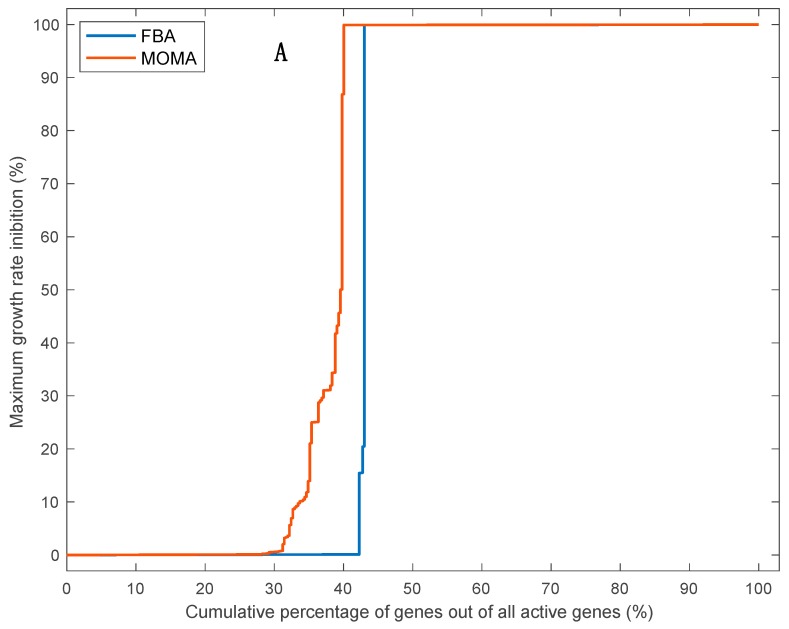
Cumulative percentage of active genes, for which deletion of each reaction (taken independently), decrease the growth rate by a proportion equal to the value on the Y-axis or lower. Calculation of intracellular flux distribution and growth rate after single gene deletion were done using flux balance analyses (FBA, blue line) or quadratic minimization of metabolic adjustment (MOMA, red line). The gene deletion analysis was performed without (**A**) or with (**B**) 53 allelic variants.

**Table 1 biology-09-00030-t001:** Summary of the 69 model parameters (names, values, units, and references). There were 66 model parameters used to determine stoichiometric coefficients of reactions in the objective functions and 3 other parameters that were fixed as constraints (C uptake rate, C:N, and N:P ratios). Additional discussion on the choices of all parameters is available in the Appendix A.

Parameters	Values	Units	Notes
Ccell	7.5	pg/cell	Measured in *F. cylindrus* (This study)
DWcell	15	pg/cell	Assuming a DWcell:Ccell ratio of 2
HCO_3_^-^ uptake	0.78	mmol g DW^−1^ h^−1^	Measured in *F. cylindrus* (This study)
C:N	5.7	mol:mol	Calculated in *F. cylindrus* (Garcia et al., 2018)
N:P	10	mol:mol	Calculated in *F. cylindrus* (Garcia et al., 2018)
Total proteins	0.46	g/g DW	Assumed (see Appendix A)
Total carbohydrates	0.31	g/g DW	Assumed (see Appendix A)
Total lipids	0.21	g/g DW	Assumed (see Appendix A)
DNA	0.0022	g/g DW	Calculated in *F. cylindrus* (This study)
RNA	0.018	g/g DW	Calculated in *F. cylindrus* (This study)
Total pigments	0.016	g/g DW	Measured in *F. cylindrus* (This study)
Chlorophyll a	0.14	pg/cell	Measured in *F. cylindrus* (This study)
Chlorophyll c	0.030	pg/cell	Measured in *F. cylindrus* (This study)
Fucoxanthin	0.060	pg/cell	Measured in *F. cylindrus* (This study)
Beta-carotene	0.0040	pg/cell	Measured in *F. cylindrus* (This study)
Diadinoxanthin	0.0090	pg/cell	Measured in *F. cylindrus* (This study)
Glucan	30	% of total carbohydrates	Assumed (see Appendix A)
Triacylglycerol	20	% of total lipids	Assumed (see Appendix A)
GC proportion	40	% of nucleobases	Calculated in *F. cylindrus* (This study)
Genome size	6.1 × 10^7^	number of bases	Calculated in *F. cylindrus* (This study)
RNA:DNA ratio	8.0	g/g DW	Based on other diatoms [27]
Glucose	0	mol/g DW	Free glucose assumed negligible
8 sugars *	0.10 to 0.32 *	mol/g DW	Measured in *P. tricornutm* [14]
20 amino acids *	0.0040 to 0.080 *	g/g DW	Measured in *P. tricornutm* [28]
19 lipid species *	0.010 to 0.12 *	mol/g DW	Measured in *P. tricornutm* [14]

* Values and names for the sugar, amino acid, and lipid species are detailed in the Appendix A, and their variations across algal species or culture conditions is discussed.

**Table 2 biology-09-00030-t002:** Summary of the cellular responses to single reaction deletions within the two sets of reactions isolated in the model (i.e., robusts or sensitives). The predicted responses using flux balance analysis (FBA) or quadratic minimization of metabolic adjustment (MOMA) included activation/inactivation of reactions and compensatory energy production through activation of photorespiration, photosystem I (PSI) cyclic electron flow, gross mitochondrial or chloroplastic ATP production, and photon harvesting. Reactions that can be taken out independently while affecting the growth rate by less than 1% are called “robust reactions”, while the reactions that abolished growth rate when deleted are deemed “sensitive reactions”, as defined in the main text.

	Robust Reactions	Sensitive Reactions
Nature of reactions	Same reaction with different energy sources	Reactions with no alternative routes
	Same reaction in different cell compartments	
Examples	5-methyltetrahydrofolate oxidoreductase using NAD or NADPH	Uptake of nitrate, phosphate, and carbon
	NADH- or ferredoxin-dependent nitrite reductase	Chrysolaminarin synthesis
	Cytosolic and mitochondrial aspartate transaminase or threonine aldolase	Biomass reactions
Reaction activation	More activated reactions than inactivated reactions	More inactivated reactions than activated reactions
Total number of active reactions	Higher for the robust reaction set than for the sensitive reaction set	
Energy dissipation	Compensatory response higher for robust reactions than for sensitive reactions	
Activation of photorespiration	Yes, for two reactions (FBA)	No activation (FBA or MOMA)
	Yes, for up to 37% of robust reactions (MOMA)	
Activation of photon harvesting	Yes, for all reactions (FBA)	Yes, for 80% of reactions (FBA)
	No activation (MOMA)	No activation (MOMA)
Activation of gross chloroplastic ATP	Yes, for all reactions (FBA)	Yes, for 80% of reactions (FBA)
	No activation (MOMA)	No activation (MOMA)
Activation of gross mitochondrial ATP	Yes, for 57% of reactions (FBA)	Yes, for 40% of reactions (FBA)
	Yes, for 100% of reactions (MOMA)	Yes, for all reactions (MOMA) but to a lesser extent
Activation of cyclic PSI	Yes, for 40% of reactions (FBA)	Yes, for 46% of reactions (FBA)
	Yes, for 60% of reactions (MOMA)	Yes, for 60% of reactions (MOMA) but to a lesser extent

## Data Availability

The final model file has been deposited in the BioModels database at https://www.ebi.ac.uk/biomodels/ with the identifier MODEL2001280001.

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
