# Peer review of "Genome-Scale Metabolic Reconstruction and in Silico Perturbation Analysis of the Polar Diatom Fragilariopsis cylindrus Predicts High Metabolic Robustness"

_biology, 2020, doi:10.3390/biology9020030_

Round 1

Reviewer 1 Report

This paper reports an in deep analysis of a Genome-scale metabolic network model in the polar diatom Fragilariopsis cylindrus. The modelling is very thorough and seems robust, even if a number of minor issues need further clarification.

Major points

That the authors do not provide their final model, at least in the form of an .sbml file. I think this is important to ensure the reproducibility of the analyses.

Linked to that, it is not entirely clear how the reactions for which no homologous gene were dealt with. Were they fully removed from the model, or were there kept without any gene sequence association?

The experimental part is described in a very concise way (one paragraph) in the SI file. Moreover, on L1356 of the Model_setup.m file on Github, a "Guérin et al., 2019" paper is mentionned. Do the authors plan to publish elswhere a more detailed version of the experimental mesurement? If so, I would recommend to pre-publish the experimental part on a preprint server, in order to get a citable doi for it.

Minor points

Regarding table 1:

Why is there a different unit for total pigment (g/gDW) and individual ones (pg/cell)? I would like to compare both, especially to see the contribution of other pigments that should be present, like violaxanthin.
Were the genome size and GC proportion really calculated from this study, or taken from Mock 2017 as stated in the SI text (L98-99) ?
Why was the RNA:DNA ration estimated based on multiple species and not based on the measured DNA and RNA contents?

L155-158: " Although no reports, to our knowledge, specifically addressed this hypothesis in eukaryotic microalgae, one may argue that natural selection, which selects optimal fitness over time, works analogously in prokaryotes and eukaryotic microalgae."
I am not convinced by this statement. See for example this 2007 paper by M. Lynch in PNAS: The frailty of adaptive hypotheses for the origins of organismal complexity (doi ?10.1073/pnas.0702207104).

L186: "high genetic divergence in F. cylindrus": high divergence relative to what?

L221: we only considered active reaction: maybe add "representing a quarter of total reactions"

SI text: why is "respiration at light" highlighted in yellow?

L4: precise "in silico " cellular perturbations?

Typos:

L42: system -> systems
L55: ditoms -> diatoms
L61: GSM means sometimes "Genome-Scale Model", sometimes "Genome-Scale Metabolic Modelling". Please be consistent.
L71: little is known on -> about?
L74: please remove the Philipps2008 citation. The only metabolic network knockout study mentionned there is the one from Deutscher2006 who is already cited separately.
L127: cylidnrus -> cylindrus
L131: detailed -> details
L134: 64 model parameters -> 66

Table 1: ref Garcia 2018 is missing from the main text (ref 4. in SI).
Ccell-> Ccell (carbon per cell)

L161: F. cylindrus,is -> F. cylindrus, is
L359: parametrizes -> parameters?
L390: equals -> equal
L421, 425: photons harvesting -> photon harvesting

Table 2: alternaive routes -> alternative
phopshate -> phosphate
Lines "Total number of active reactions" and "Energy dissipation": please put the text in the middle of both columns, to make sure that the reader understands the statement is for both reaction types
photons harvesting -> photon harvesting

L533: a proportion equals to -> equal
L542: system biology -> systems
L544: genetic divergence increase -> increases

Table S1:
L2: Reaction name -> Protein name
CAMT_c Cycloartenol -> Cycloartenol methyltransferase?

Comment on Github files:

Script localsens_script_vf.m: what is it for? It is not mentionned in the article or in the readme. Please explain or delete it.

Script ReactionDeletion_sensitivityRxn.m is presumably aluded in the readme as 'ReactionDeletionSensitivity.m'. Please check and homogenize names.

Script analyse_reseau_2.py:
Starting from L113, all code comments are in french. Please translate into english.

Reviewer 2 Report

Comments

The authors investigated metabolism in a polar diatom, Fragilariopsis cylindrus through developing genome-scale metabolic model. The authors tried to underscore the strong robustness of metabolism in F. cylindrus. The study appeared original. While the authors generated some physiology data for modeling of F. cylindrus, information on growth conditions is incomplete even in the supplementary file.  It would be good to have experimental validation of some major predictions. Also, more discussions should be provided on the genome-scale modeling. The authors should address the comments below carefully and soundly prior to publication.

Major comments:

The title of the manuscript may be improved to be more concise and precise. Genome sequence information of the diatom such as number of encoding genes may be covered briefly for better discussions. In Table 1, it’s not quite clear how the values such as pigment content were measured and calculated. How many replicates were used? In Fig.1, it seems change of pigments such as chlorophyll a and major carotenoids didn’t affect growth rate. Any clear explanations? In Fig.2, the upper and lower panels show the same data with different scales. Overall, the figure should be improved. In addition, some representative parameters may be indicated or highlighted. Table 2 may be better organized. In Fig.4, relative frequency may be visualized better. In all 69 model parameters, it seems only carbon uptake rate significantly affects growth rate. Is that because there is a correlation between carbon uptake and biomass generation rate? The light wavelength may affect photosynthesis efficiency. Did authors take into account any light-driven metabolism? In general, the limitation of the modeling in this study should be discussed as there are many assumptions without direct physiology data.

Other comments:

The Conclusions section is a bit vague and the development of genome-scale metabolic model and its implications may be described more clearly. Is there any calibration curve for the measurements (e.g. pigments)?

Round 2

Reviewer 1 Report

The authors responded to all request in a reasonable way. The paper can be published now as it is.

Reviewer 2 Report

I think the authors have addressed the issues I raised.